# Comparison of Bone Marrow Biopsy and Flow Cytometry in Demonstrating Bone Marrow Metastasis of Neuroblastoma

**DOI:** 10.3390/diagnostics14242776

**Published:** 2024-12-11

**Authors:** Esra Arslantaş, Ali Ayçiçek, Selvinaz Özkara, Ayşe Özkan Karagenç, Sibel Akpınar Tekgündüz, Duygu Yıldırgan, Tuba Nur Tahtakesen Güçer, Ayşe Gonca Kaçar, Özgü Hançerli, Saide Ertürk, Ezgi Paslı Uysalol, Cengiz Bayram

**Affiliations:** 1Department of Pediatric Hematology and Oncology, Başakşehir Çam and Sakura City Hospital, Istanbul 34480, Turkey; ayciceka@hotmail.com (A.A.); ayseozkankaragenc@gmail.com (A.Ö.K.); drsibel76@gmail.com (S.A.T.); duyguozkorucu@gmail.com (D.Y.); tuuba-nur@hotmail.com (T.N.T.G.); goncakacar@gmail.com (A.G.K.); drozgu@hotmail.com (Ö.H.); saideerturk@gmail.com (S.E.); ezgipasli@yahoo.com (E.P.U.); cengizbayram2013@gmail.com (C.B.); 2Department of Pathology, Başakşehir Çam and Sakura City Hospital, Istanbul 34480, Turkey; selvinazo@gmail.com

**Keywords:** neuroblastoma, children, metastasis, bone marrow, bone marrow biopsy, flow cytometry

## Abstract

Objective: This study aimed to compare bone marrow aspirate (BMA) multicolor flow cytometry (MFC) analysis and bone marrow biopsy (BMB) in detecting bone marrow (BM) involvement in children with neuroblastoma (NB) at diagnosis and during follow-up. Materials and Methods: A total of 132 BM samples from 39 patients (M/F ratio: 19/20; median age: 38 months) with neuroblastoma were simultaneously obtained for evaluation. The samples were investigated for BM involvement using BMB and MFC. Results: A comparison between MFC (*n*: 60) and BMB (*n*: 60) was possible for 120 samples. When BMB was considered as the reference standard, MFC had diagnostic sensitivity, specificity, positive predictive value, and negative predictive value of 86%, 58%, 54%, and 88%, respectively, and values of 90%, 57%, 60%, and 89%, respectively, at diagnosis. The median proportion of CD45−/CD56+ cells in MFC was 0.028% (range 0–35%). The event-free survival (EFS) rates for MFC (+) and MFC (−) patients according to the analysis results of the BM samples at the time of diagnosis were 70.6% and 81.8%, respectively (*p* = 0.607), and the overall survival (OS) rates were 88.2% in MFC (+) patients and 90.9% in MFC (−) patients (*p* = 0.583). Conclusion: Multicolor flow cytometry may be used as an adjunct to cytomorphology to achieve more sensitive and accurate results as an objective, quantitative method with fast results in detecting bone marrow involvement in children with NB.

## 1. Introduction

Neuroblastoma (NB) is the most prevalent extracranial solid tumor in children and can arise from various locations within the sympathetic nervous system [1]. The etiology of NB has not been completely elucidated yet, but it may be related to maternal, peri-, and post-natal exposures to risk factors or adverse gestational events [2,3,4,5,6]. Due to the widespread distribution of the sympathetic nervous system throughout the body, neuroblastoma can develop in various locations. It most often originates in the abdomen, with the adrenal glands being the most common primary site. Cervical and thoracic involvement is mostly observed in infants. Invasion into the orbital bones and ensuing the development of so-called ‘racoon’ eyes are known to be characteristic findings for NB [7]. The International Neuroblastoma Risk Group Staging System (INRGSS) is currently used globally for the staging and determination of risk groups [8,9]. Demonstration of metastasis is of vital importance in determining the stage of the disease, risk groups, and, consequently, the treatment protocol and prognosis. Bone marrow (BM) is the most common site of metastases, and up to 50% of patients have BM involvement at diagnosis; the rate of bone marrow involvement may increase to 80% in patients with high risk or metastatic disease [10,11]. The detection of bone marrow involvement plays an important role in determining the risk group of the disease and thus in choosing the treatment protocol. The synchronous evaluation of BM biopsy specimens and bone marrow aspirate has been reported as the most accurate method for detecting BM metastases, in line with the recommendations of the International Neuroblastoma Risk Group, and the combination of BMB and aspirates was noted to be associated with a high sensitivity of 94.7% [12,13]. However, it is difficult to obtain trephine BM biopsy material, especially in infants and young children, and it is also known that sampling from the tumor becomes difficult if, as usual, less than 30% of BM involvement occurred due to multilocular involvement in BM. The search for a more sensitive and more reliable method for evaluating BM has led to the use of methods such as multicolor flow cytometry (MFC), immunocytologic analyses of BM sections, and quantitative polymerase chain reaction (PCR) [14,15]. Although not yet in routine use, MFC has been described as a rapid and useful method to complement standard histopathological methods for the diagnostic evaluation and detection of BM micrometastases of pediatric solid tumors, including neuroblastoma [15,16]. Neuroblasts are known to express the intensity of the neural cell adhesion molecule (N-CAM), also known as CD56, and they do not express CD45, a marker for common leukocyte antigens, typically expressed on hematopoietic cells [17]. This phenotypic feature of neuroblasts can be used for demonstrating bone marrow metastasis of neuroblastoma.

The aim of this study was to compare the results of MFC bone marrow aspirate analysis with BMB in the detection of neuroblasts during the diagnostic process and follow-up of patients with NB and to investigate the utility of MFC results in terms of disease prognosis.

## 2. Materials and Methods

### 2.1. Study Design and Sample Size

Eligible patients (*n*:39) with newly diagnosed children (0–18 years of age) with neuroblastoma were included between May 2020 and September 2023 at İstanbul Başakşehir Çam and Sakura City Hospital, Istanbul, Turkey. Bone marrow biopsy and 0.3–1 mL BM aspiration (BMA) through the same access point at the diagnosis and during the follow-up period were performed in all patients. Of the 39 (F/M ratio: 19/20) patients included, 48 BM samples were collected at the time of diagnosis, and 84 samples were collected during the follow-up. After excluding the non-diagnostic BMB results (due to insufficient material), it was possible to compare BMB and simultaneous MFC results in 120 BM samples.

### 2.2. Patients

Demographic characteristics, histopathological subgroups, disease stages, risk groups, overall survival, and event-free survival data of the patients were evaluated. The INRGSS was used to determine the disease stage and risk groups of the patients. Risk groups were determined according to disease stage, age, and primary tumor characteristics, including histopathological category, degree of differentiation, ploidy status, and presence of N-myc amplification and 11q aberration. The decisive criteria for inclusion in HRG were the presence of N-myc amplification and metastasis, and N-myc amplification was examined using the Fluorescence In Situ Hybridization (FISH) method on fresh tissue traces or formalin-fixed paraffin-embedded tissue blocks. The patients were treated using the Turkish Pediatric Oncology Group (TPOG)’s Neuroblastoma Protocols (2009 and 2020).

### 2.3. Evaluation of Bone Marrow Samples

Bone marrow biopsies were embedded in paraffin blocks after fixation and acid decalcification and underwent routine tissue tracing procedures. The sections were cut to two microns stained with hematoxylin and eosin (H&E), and immunohistochemistry (IHC) was applied to sections. Tissue examinations were performed to detect tumor foci in consecutive repetitive serial sections, and IHC was added to highly suspicious areas. The slides were then transferred into the Ventana Autostainer Bench Mark ULTRA (Ventana Medical systems, Tucson, AZ, USA) and stained with MPO, CD56, synaptophysin, and chromogranin. After antigen retrieval, slides were stained with myeloperoxidase Rabbit Polyclonal Antibody (Cell Marque, Rocklin, CA, USA), CD56 (MRQ-42, Cell Marque, Rocklin, CA, USA), synaptophysin (MRQ-40, Cell Marque, Rocklin, CA, USA) and chromogranin A (clone LK2H10, Ventana Medical systems) as the primary antibodies for 40, 36, 68, and 32 min, respectively. Formalin-fixed, paraffin-embedded (FFPE) tissue sections with known positivity for antibodies were used as positive controls and to compare the staining intensity. Immuno-stained slides were evaluated under a light microscope. The presence of metastasis in the BMB specimens was evaluated by a histopathologist who was blinded to the flow cytometry results. Figure 1a–f show an example BMB sample infiltrated with NB cells.

Samples of BMA for MFC analysis were collected in tubes containing anticoagulant ethylenediaminetetraacetic acid (EDTA), and they were delivered to the laboratory within two hours of collection. Aspirates smeared on slides and stained with May–Grünwald Giemsa (MGG) were investigated for NB cells or clusters by at least two hemato-oncologists using light microscopy (BMA-LM). The panel of combinations used in MFC was cMPO/CD3/cCD3/CD19/c79a/cTDT/CD33/CD34/CD45 plus CD56, and immunophenotyping at diagnosis was conducted, with 30,000 events being collected. Analysis of the BMA MFC data was performed using Kaluza software (Beckman Coulter, Brea, CA, USA). The Navios EX device set-up was controlled with Stem-Trol™ Control Cells (Beckman Coulter). An amount of 5 µL of monoclonal antibodies was added to 50 µL of BM aspirates. Cell surface antibodies were first added to the BM samples and incubated for 15 min. Then, an equal volume of IntraPrep Solution Buffer 1 (fixative reagent) was added, followed by a second incubation period for 15 min. Next, 3 mL of IsoFlow Sheath Fluid was added, and the sample was centrifuged at 300× *g* for 5 min. After separating the supernatant, IntraPrep Reagent 2 (permeabilization solution) was added, and the sample was incubated for 5 min. In the next step, cytoplasmic monoclonal antibodies were added, followed by a 15 min incubation period. Finally, 3 mL of IsoFlow Sheath Fluid was added, the sample was centrifuged again, and the supernatant was separated, and 500 µL of IsoFlow Sheath Fluid was added. CD45−/CD56+ cells were gated for identification of NB cells. At each stage, mixtures were made using a vortex device. Detection of >0.01% neuroblasts among all nucleated cells that were also gated on the forward scatter and side scatter (FSC/SSC) plot by MFC in BMA samples was considered MFC (+) [15] (Figure 2).

### 2.4. Primary and Secondary Outcomes

Presence of tumor infiltration in BMB specimens was indicated as BMB (+), and detection of >0.01% neuroblasts among all nucleated cells as gated on the forward scatter and side scatter (FSC/SSC) plot by MFC in BMA samples was considered MFC (+). The concordance between the results for two methods and the sensitivity and specificity of MFC in demonstrating BMI were calculated. Overall survival (OS) was defined as the time from diagnosis to death, and event-free survival (EFS) was defined as the time from diagnosis to first relapse or primary progressive/refractory disease or death. The impact of MFC results on survival rates was investigated.

### 2.5. Statistical Analysis

The data were analyzed using SPSS for Windows, version 22.0 (IBM Inc., Armonk, NY, USA). Descriptive statistics were employed to analyze demographic and clinical data. Continuous data with a normal distribution were presented as mean ± standard deviation (SD), while data with a non-normal distribution were presented as median (range: minimum–maximum). Categorical variables were presented as numbers and percentages. BMB and MFC data were analyzed in terms of diagnostic sensitivity, specificity, and positive (PPV) and negative (NPV) predictive values for the assessment of BM involvement. Event-free survival (EFS) and overall survival (OS) curves were estimated and plotted using the Kaplan–Meier method, and the log-rank test was used to compare survival differences between groups. *p* < 0.05 was considered statistically significant.

## 3. Results

Thirty-nine patients (male/female ratio: 19/20) with NB who simultaneously underwent BMB and MFC analyses of BM aspirates were included. The median age at diagnosis was 38 months (range of 4–95 months). The identified histopathological subgroups were neuroblastoma in 33 patients (84.6%) and ganglioneuroblastoma in 6 patients (15.4%). Of the 39 patients with neuroblastoma, N-myc was studied in 36. In 13 of these patients, the tumors were N-myc positive, and N-myc amplification could not be performed in 3/39 patients due to insufficient material. Throughout the study period, four patients had a pre-existing diagnosis of NB and were undergoing treatment, and one of these patients had recurrent disease, while the remaining thirty-five patients were newly diagnosed. There were 7 patients (18%) in the low-and-very-low-risk group, 4 patients (10%) in the intermediate-risk group, and 28 patients (72%) in the high-risk group. Twenty-five patients (64%) had metastatic disease. The primary tumor sites in order of frequency were the abdomen (67%); chest (26%); inguinal region (5%); and cervical region (2%). At the diagnosis, the mean ± SD values for white blood cell (WBC) count were 10,031 ± 2979/mm^3^; for hemoglobin concentration, the values were 9.5 ± 1.8 g/dL; for platelet count, the values were 369 ± 146 10^9^/L; the median value for lactate dehydrogenase (LDH) was 518 U/L (range 214–7017); and for neuron-specific enolase (NSE), the value was 189 (range 11.0–2307) ng/mL. During the study period, 11 (28%) patients developed relapse or tumor progression, while 6 patients (15%) died. The patients were followed up with for a median of 19 months (range of 5–40 months).

A total of 132 samples (66 BMB and 66 MFC samples) were obtained simultaneously, and 56 (28 BMB and 28 MFC samples) of the 132 samples were obtained at the time of diagnosis. Among all samples, BM involvement was detected in 39/66 samples by MFC and in 22/66 samples by BMB. When the BM samples obtained at the time of diagnosis were analyzed separately, BM involvement was found in 15/28 patients by an MFC analysis and in 10/28 patients by BMB. The median proportion of CD45−/CD56+ cells on MFC was 0.028% (range of 0–35%).

Six BMB samples were found to be inadequate for re-evaluation, so a comparison between MFC and BMB was possible in 60 BMB and 60 MFC samples, and 41/60 (68.3%) of the results were concordant. As 4/28 (14.3%) BMB samples were not eligible for evaluation, 24 BMB and 24 MFC samples obtained at the time of diagnosis were compared. In this subgroup, 17/24 (70.8%) results were concordant (Figure 3a). Five of six patients who were reported as MFC (+) and BMB (−) at diagnosis were already included in the high-risk group because they already had metastatic disease (outside the BM) and/or N-myc positivity. The one remaining patient did not have an advanced disease stage and was not included in the high-risk group, and no relapse or death was observed in the 40-month follow-up. A comparison of positive results by MFC, BMA-LM preparations, and BMB sections is shown in Figure 3b.

When BMB was considered as the reference standard, MFC had diagnostic sensitivity, specificity, PPV, and NPV of 86%, 58%, 54%, and 88%, respectively, and these values were 90%, 57%, 60%, and 89%, respectively, at the time of diagnosis (see Table 1).

Based on the samples obtained at the time of diagnosis, the EFS rates for BMA MFC (+) and BMA MFC (−) patients were 70.6% and 81.8%, respectively (*p* = 0.607), and the OS rates were 88.2% for BMA MFC (+) and 90.9% for BMA MFC (−) patients (*p* = 0.583) (Figure 4a,b).

## 4. Discussion

MFC is currently an essential method for both rapid diagnosis and classification and for the measurement of treatment responses in hematologic malignancies, especially leukemias; however, it is not yet routinely used to demonstrate BM involvement in solid tumors [10]. When NB is not diagnosed by BMA examination but is investigated in patients with an NB diagnosis that is both suspected regarding hemato-oncology and histopathologically confirmed by biopsy, confidence in these MFC findings of CD45−/CD56+ cells is increased. Neuroblasts are characterized by the presence of a neural cell adhesion molecule (NCAM aka CD56), which can be detected by appropriate monoclonal antibodies, and the absence of any isoform of the common leukocyte antigen, CD45. CD56 is actually a cell membrane glycoprotein that contains immunoglobulin-like domains and is highly expressed in nerve cells, various neuroendocrine tumors, and tumor-infiltrating lymphocytes. While it plays a crucial role in the growth and development of the nervous system, helping to promote cell-to-cell adhesion, it can also affect the infiltration, invasion, and metastasis of malignant tumors [18]. Leukocyte common antigen 45 (CD45), another membrane glycoprotein typically expressed on hematopoietic cells, is used in the MFC assay to distinguish neuroblasts from hematopoietic cells. In studies investigating MFC in the detection of BM infiltration of NB, in addition to the CD45−/CD56+ phenotype, CD81+, GD2+, NB84+, CD9+, CD90+, and CD73− phenotype cells were also gated, and BM involvement was investigated [10,13,15,16,17,19]. The CD45−/CD56+ phenotype is not specific only to neuroblasts; it can also be observed in other solid tumor types [17,20,21]. While there is a CD99+/CD45− phenotype in Ewing sarcoma, CD45−, CD90+, CD56+, and CD57−/+ phenotypes are most frequently observed in rhabdomyosarcoma. Primitive neuroectodermal tumors (PNET) coexpress CD57, CD99, and CD56 [16,22]. Although these phenotypes have been described, a proven comprehensive biomarker panel for the diagnostic screening of pediatric solid tumors has not yet been established.

In the current study, we examined the presence of CD45− and CD56+ cells in BMA samples by MFC, and we defined BMA samples as MFC positive if CD45−/CD56+ cells represented more than 0.01% among all nucleated cells gated on the FC/SSC plot [15]. All of the patients included in this study had a histopathologically confirmed diagnosis of NB. We compared the results of MFC with those of BMB and found that 68% of the results were concordant for both methods, and when BMB was considered as the reference standard, MFC had 86% sensitivity, 58%, specificity, 54% PPV, and 88% NPV. Manenq C et al., in a similar study, examined both follow-up and diagnostic BMB and MFC results in 21 patients. The results were concordant in 26/31 (89%) patients, while at the time of diagnosis, all results were concordant. They reported a sensitivity for MFC of 69.2%, specificity of 94.4%, PPV of 90%, and NPV of 81% in relation to the BMB reference standard. They reported that the combination of MFC and BMA-LM helped detect BM disease with higher specificity (94.4% vs. 77.8%) and sensitivity (76.9% vs. 61.5%) compared to BMA-LM alone. These authors suggested that BMA-LM and MFC, when used together, provide rapid and satisfactory results, which can enable rapid intervention by detecting BM metastasis, especially in emergency situations [23]. Jain et al. compared the results of MFC, BMA-LM, and biopsy samples taken both at the diagnosis and follow-up in patients with NB [13]. The authors found 72% concordance among negative results across all three methods, while 61% concordance was reported among positive results across all three methods. In 16% of the samples, only MFC positivity was detected (either or both cytology and histology were negative), while positive results for both cytological and histological methods were obtained in 23% of the samples in which MFC was negative. In conclusion, Jain et al. argued that MFC disclosed low-level residual disease that could not be detected cytologically or histologically and that MFC was an objective and quantitative diagnostic method. In the present study, 14/60 (23.3%) samples were positive for all three methods (BMB, BMA-LM, and MFC), and 22/60 (36.7%) were negative for all three methods.

Flow cytometry and histopathological comparisons in patients diagnosed with NB have been made in tissues and body fluids as well as in BM. In a study that included BMB, peripheral blood, primary tumor, hepatic nodule, and rib fragment samples, FCM and histopathological methods were found to be consistent in 88.9% of the cases; when the histopathological method was considered as the standard reference, MFC had diagnostic sensitivity, specificity, PPV, and NPV of 100%, 86%, 67%, and 100%, respectively [24]. This study also took into account the turnaround time of MFC and reported a significantly faster yield with MFC, with a median of 30 h and 45 min with MFC compared to 94 h and 49 min with the pathological examination/immunohistochemistry method. The authors argued that MFC is an acceptable, fast, and safe method for the diagnosis and follow-up of children with NB, and it would be useful to add it to conventional methods, where possible. In another study in which pleural and peritoneal effusion samples of patients diagnosed with NB were compared using MFC and cytomorphology, the authors reported that there were no significant differences between MFC and cytomorphology, and they suggested that MFC may be used as an auxiliary method to cytomorphological methods in the detection of neuroblasts in effusion fluid samples [10].

While assessing the importance and effectiveness of MFC in demonstrating BM involvement in NB, an important question that should be answered is whether MFC has an effect on the prognosis of the disease. It is instructive to compare the treatment outcomes of MFC-positive and -negative patients. Although it is difficult to detect minimal residual disease (MRD) by MFC in neuroblastoma, which is among the CD45 negative non-hematopoietic neoplasms, it has been reported that initial bone marrow investigation by MFC has an important prognostic value. The prognostic significance of the detection of neuroblasts was investigated using MFC in initial BM samples in 51 patients with NB, and it was reported that MFC detected greater percentages of neuroblastoma cells than the conventional cytomorphologic method (49% vs. 29.4%; *p* = 0.043) and that MFC-positive patients had significantly worse outcomes than MFC-negative patients, with worse EFS and cumulative incidence of relapse/progression. Popov et al. also compared MFC with real-time quantitative reverse transcriptase polymerase chain reaction (RQ-PCR) in a subset of patients with neuroblastoma, and they argued that the detection of BM involvement by MFC was even more robust than RQ-PCR and thus has a very strong prognostic impact [15]. A similarly designed study including 27 patients with NB was conducted, and the researchers reported that patients with <0.01% or the absence of NB cells detected by MFC had significantly better outcomes than others based on an OS analysis. It was reported that the use of MFC, which is a fast and reliable method, together with conventional cytomorphologic assays, both at diagnosis and during the follow-up, can provide more accurate results [17]. Similarly, in the present study, OS and EFS tended to be better in MFC-negative patients than in MFC-positive patients (*p* > 0.05); a larger sample size with a longer follow-up may yield different, possibly significant results.

## 5. Limitations

This study has several limitations. Since it was a single-center and relatively short-term study, a limited number of patients and bone marrow samples were included. A longer follow-up period may more clearly reveal the relationship between MFC results and disease prognosis.

## 6. Conclusions

As an objective, quantitative method with fast results, MFC offers good diagnostic sensitivity and specificity in the detection of bone marrow involvement in pediatric patients diagnosed with NB. We recommend the use of MFC in addition to cytomorphologic methods in order to detect cases with a small area or diffusely scattered neuroblasts in BM samples, which are challenging to obtain in young children. Furthermore, these scarce cells may be overlooked when histopathological or cytomorphological methods are used in order to improve the diagnosis of NB and more effectively assess the risk for optimal treatment. For more definitive results, more studies with a higher number of patients are needed.

## Figures and Tables

**Figure 1 diagnostics-14-02776-f001:**
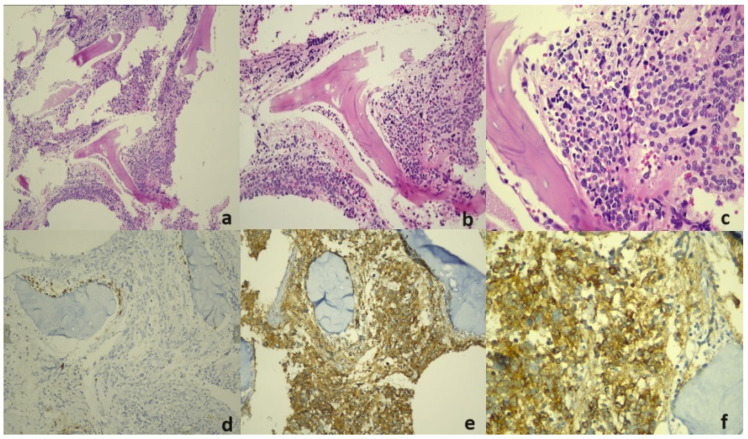
(**a**) Bone marrow biopsy showing infiltration of neuroblastoma cells that largely eliminate bone marrow parenchymal elements in diffuse pattern (H&Ex100). (**b**,**c**) Bone marrow biopsy showing neuroblastoma cells with oval, round nucleus and thin chromatin containing small nucleoli (A: H&Ex200; B: H&Ex400). (**d**) Bone marrow biopsy showing small number of borderline myeloid series elements (IHCX200) in paratrabecular area with myeloperoxidase (MPO). (**e**,**f**) Chromogranin-positive neuroblastoma cells infiltrating bone marrow (A: IHCX200; B: IHCX400).

**Figure 2 diagnostics-14-02776-f002:**
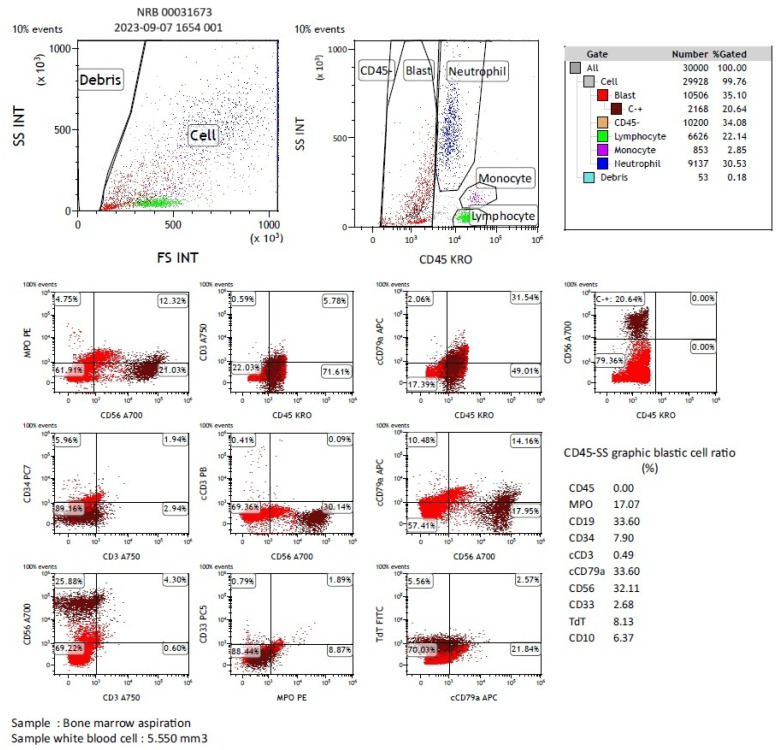
Dot plot diagrams indicating neuroblastoma cells in bone marrow samples by flow cytometry. Clusters in the CD45/CD56 dot plots C−+ gate correspond to metastatic neuroblastoma cells (CD45−/CD56+). Monoclonal antibodies are listed in the legend of each dot plot. Nonviable cells were excluded based on their lower forward and sideward light scatter features (upper-left dot plot). Of note, events require a mean fluorescence intensity of >10^4^ to be considered positive.

**Figure 3 diagnostics-14-02776-f003:**
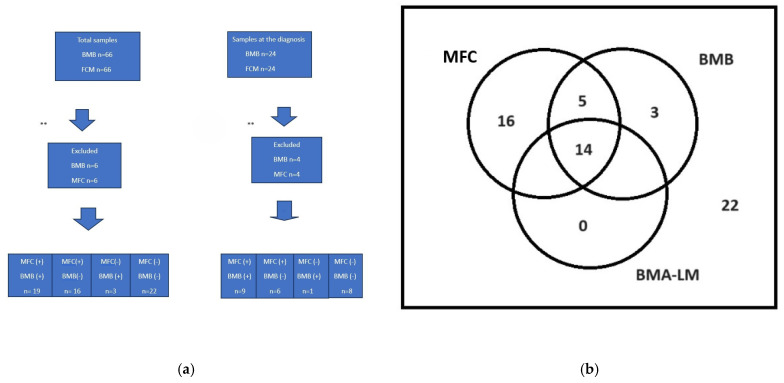
(**a**) Comparison of bone marrow biopsy and multicolor flow cytometry results, all results, and results obtained at time of diagnosis. ** Unidentified BMB results (and, simultaneously, MFC results) were excluded when comparing both methods. (**b**) Comparison of results by MFC, cytomorphological examination of BMA, and BMB. Numbers in circles represent positive results after excluding non-diagnostic BMB and simultaneous MFC and BMA morphological evaluations. Number outside of circles represents negative results for MFC, BMB, and BMA-LM.

**Figure 4 diagnostics-14-02776-f004:**
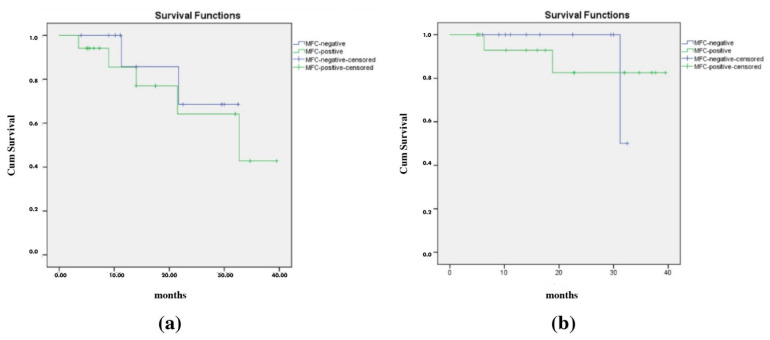
(**a**) Comparison of event-free survival curves of patients with MFC-positive and MFC-negative results. (**b**) Comparison of overall survival curves of patients with MFC-positive and MFC-negative results.

**Table 1 diagnostics-14-02776-t001:** Diagnostic values for flow cytometry.

	Sensitivity	Specificity	PPV	NPV
MFC	86	58	54	88
MFC *	90	57	60	89

MFC: multicolor flow cytometry (all samples)—when bone marrow biopsy is considered as reference standard. MFC *: multicolor flow cytometry (at diagnosis)—when bone marrow biopsy is considered as reference standard.

## Data Availability

All data used in this study can be obtained from the author upon request.

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
