# Peer review of "Comparison of Bone Marrow Biopsy and Flow Cytometry in Demonstrating Bone Marrow Metastasis of Neuroblastoma"

_diagnostics, 2024, doi:10.3390/diagnostics14242776_

Round 1
Reviewer 1 Report
Comments and Suggestions for Authors
An interesting and important research is devoted to study of the multicolor flow cytometry (MFC) as an auxiliary method to complement bone marrow aspirate (BMA) and bone marrow biopsy (BMB) analysis techniques for neuroblastoma diagnosis and potential use of the MFC results for the prognosis of the disease.
The material is logically structured. The scheme of the study design is provided, which is very illustrative and convenient.
Authors also describe the limitations of the study which is a benefit.
The following comments do not diminish the value of the Article:
Please include the heading to the Abstract (Background/Objectives, Methods, Results, and Conclusions), according to the Instructions published on the Journal’s website.
The main markers studied are characterized (CD56, CD45), would you please also describe a bit in more details the panel of combinations used in MFC (CD3/cyCD3/CD7/CD19/TDT/CD34).
Line 16 Would you please decipher the following abbreviatures: PPV and NPV.
Line 43-45 Would you please describe a bit the specificity and the benefits of the bone marrow biopsy (BMB), bone marrow aspirate (BMA)analysis techniques?
Line 50 Please define the MFC acronym at the first time it appears in the main text of the Article.
Line 66 Would you please name the Center where the research had been carried out.
In the Line 65 it is indicated that there were 39 patients, and in the Lines 67-68 there is also information about 48 and 84 patients involved. Would you please specify?
Line 69 Would you please explain a bit the specificity of nondiagnostic BMB results, or maybe please provide the reference.
Lines 79-80 Would it be possible to describe a bit more in details recorded parameters: ‘Demographic characteristics, disease stages, risk groups and survival data of the patients’.
Lines 84-85 Would you please specify a bit immunohistochemistry (IHC panel) technique that was used during the study.
Line 93 Figure 1 Would it be possible to add also scale bars to the figures.
Lines 97-99 Would you please define acronyms IHK, MPO.
Line 124 Would you please specify a bit more in details how the sensitivity and specificity of MFC were calculated.
Line 132 Would you please specify the range for continuous data presentation.
Lines 134-135 Would you please describe how the positive (PPV) and negative (NPV) predictive values for the assessment of BM involvement were calculated?
Line 144 Would you please characterize a bit analysis of the N-myc.
Lines 167-168 Would you please explain a bit when the MFC and BMB results are considered as concordant.
Line 170 Would you please explain a bit more in details in the Materials and methods section the specificity of the division of studied population into the subgroups.
Line 172 Would you please describe a bit more in details the criterion of inclusion the patients into high risk group?
Line 186 Would you please specify what does the number outside the circles mean?
Line 257 Would you please define the acronym ‘PA/IHC’ method.
Please check for misspellings
Please describe the references according to the Journal’s requirements.
Author Response
Dear Editor,
Thank you for the evaluation of our manuscript and the suggestions by the reviewers. Please find below the revised version of our manuscript together with our responses to the reviewers.
Kind regards,
Dr. Esra ArslantaÅŸ
Responses to Reviewers;
Reviewer 1:
- Please include the heading to the Abstract (Background/Objectives, Methods, Results, and Conclusions), according to the Instructions published on the Journal’s website.
The abstract section is reorganized with the addition of the heading according to the instructions published on the Journal's website.
- The main markers studied are characterized (CD56, CD45), would you please also describe a bit in more details the panel of combinations used in MFC (CD3/cyCD3/CD7/CD19/TDT/CD34).
Detailed information about flow cytometry and panels is added and incorporated into the '2.3.Evaluation of Bone Marrow Samples' section."
- Line 16 Would you please decipher the following abbreviatures: PPV and NPV
The abbreviations PPV and NPV is added in the manuscript.
- Line 43-45 Would you please describe a bit the specificity and the benefits of the bone marrow biopsy (BMB), bone marrow aspirate (BMA)analysis technique
A sensitivity value for detecting bone marrow invasion using both BMB and BMA analysis is added to this section.
- Line 50. Please define the MFC acronym at the first time it appears in the main text of the Article.
The acronym MFC is defined in the main text in this paragraph.
- Line 66 Would you please name the Center where the research had been carried out.
The name of the center where the research had been carried out is added to this section.
- In the Line 65 it is indicated that there were 39 patients, and in the Lines 67-68 there is also information about 48 and 84 patients involved. Would you please specify?
There are 48 BM samples at diagnosis and 83 samples during follow-up from 39 patients. This was clarified in the manuscript.
- Line 69 Would you please explain a bit the specificity of nondiagnostic BMB results, or maybe please provide the reference.
The results referred to as 'nondiagnostic BMB results' were biopsy results that ended up as insufficient material from the pathology department for involvement of BM of neuroblastoma. This was clarified in the manuscript.
- Lines 79-80 Would it be possible to describe a bit more details recorded parameters: ‘Demographic characteristics, disease stages, risk groups and survival data of the patients’.
Demographic characteristics (age at the diagnosis, gender), histopathological subgroups (neuroblastoma, ganglioneuroblastoma), disease stages, risk groups and overall survival and event free survival data of the patients were detailed.
- Would you please specify a bit immunohistochemistry (IHC panel) technique that was used during the study.
A detailed paragraph for the Immunohistochemistry (IHC panel) technique has been added to the '2.3.Evaluation of Bone Marrow Samples' section.
- Line 93 Figure 1 Would it be possible to add also scale bars to the figures
Unfortunately this requires repeated photography of the samples and was not possible technically at the time of revision.
- Lines97-99 Would you please define acronyms IHK, MPO.
Definition of acronyms were added.
- Line 124 Would you please specify a bit more in details how the sensitivity and specificity of MFC were calculated.
The sensitivity and specificity of MFC were calculated using the formulas below:
True positive rate (TPR), sensitivity, Recall, probability of
Detection= ∑ True positive / ∑condition positive
False positive rate (FPR), Fall-out, probability of false
Alarm= ∑ False positive / ∑condition negative
False negative rate (FNR), Miss rate,
∑ False negative / ∑condition positive
True negative rate (TNR), Specificity
∑ True negative / ∑condition negative
- Line 132 Would you please specify the range for continuous data presentation.
In the section ‘ Statistical Analysis’ , the sentences regarding continuous data presentation has been detailed.
- Lines 134-135 Would you please describe how the positive (PPV) and negative (NPV) predictive values for the assessment of BM involvement were calculated?
Along with the statistical formulas in the 13th comment, PPV and NPV calculated with additional formulas
Positive likehood ratio =TPR/FPR
Negative likhood ratio= FNR/TNR
- Would you please characterize a bit analysis of the N-myc.
Additional methods regarding N myc analysis are added to the section
‘2.2 Patients ‘.
- Lines 167-168 Would you please explain a bit when the MFC and BMB results are considered as concordant.
Concordant and discordant results are detailed in figure 3a in the manuscript.
- Line 170 Would you please explain a bit more in details in the Materials and methods section the specificity of the division of studied population into the subgroups.
Material and methods were detailed and reorganized.
- Line 172 Would you please describe a bit more in details the criterion of inclusion the patients into high risk group?
This information was added in the material and methods section.
- Line 186 Would you please specify what does the number outside the circles mean?
This was specified.
- Line 257 Would you please define the acronym ‘PA/IHC’ method.
This was defined.
- Please check for misspellings.
This was re-checked.
- Please describe the references according to the Journal’s requirements.
This was re-checked.
.
Reviewer 2 Report
Comments and Suggestions for Authors
The authors retrospectively analyzed the data of flow cytometry, aspiration, biopsy of bone marrow samples of patients with neuroblastoma. Based on biopsy results as the gold diagnostic tool, they reported the diagnositc performance (sensitivity, specificity, positive and negative predicative values) of flow cytometry. This topic is interesting and the dataset potentially important in further defining the role of flow cytometry in diagnosis of bone marrow metastasis of neuroblastoma and perhaps some other solid tumors. This article is well written. The materials and data are clearly presented.
I have some comments and suggestions.
1#. The authors defined MFC positive if CD 45(-) CD 56 (+) 224 cells represented more than 0.01% among all nucleated cells. Is there a reference or similar standard for this definition. Some definitions are CD15-/CD56+/GD2+. The cutoff is very important and needs to be well defined. 0.01% appears very likely to be affected by contaminations.
2# The authors present OS for flow cytometry positive and negative patients. They should present other clinical data of this population to support the accuracy of flow cytometry. For example, the stage, tumor burden of all patients. It is especially important to report those who had discordant results between flow cytometry and biopsy reports. Do they have have early stage disease and therefore, overdiagnosis by flow cytometry (MFC+BMB-) ? or they actually have advanced stage so bone marrow metastasis is very likely and only detected by flow cytometry?
3# The sample processing methods should also be described. Cancer cells may be lost either by processing the biopsy or flow cytometry samples and thereby affecting the accuracy of diagnostic tools.
4# In total samples, N=16 (MFC+BMB-). Most of such samples are follow up samples so they are likely to have minimal residual disease, if MFC correctly detected the disease. It is very interesting. The authors should analyze the clinical outcome of this group in comparison with the MFC-BMB- group and MFC+BMB+ group. If the outcome is similar to the MFC+BMB+ group then MFC is a good tool in diagnosis of residual disease in marrow.
Author Response
03.12.24
Dear Editor,
Thank you for the evaluation of our manuscript and the suggestions by the reviewers. Please find below the revised version of our manuscript together with our responses to the reviewers.
Kind regards,
Dr. Esra ArslantaÅŸ
Responses to Reviewers;
Reviewer II
1#. The authors defined MFC positive if CD 45(-) CD 56 (+) 224 cells represented more than 0.01% among all nucleated cells. Is there a reference or similar standard for this definition. Some definitions are CD15-/CD56+/GD2+. The cutoff is very important and needs to be well defined. 0.01% appears very likely to be affected by contaminations.
1#. There is limited data on MFC of children with neuroblastoma. A similiar study by Popov et al defined %0.01 as a standart cut off. (Popov A, Druy A, Shorikov E, Verzhbitskaya T, Solodovnikov A, Saveliev L, Tytgat GAM, Tsaur G, Fechina L. Prognostic value of initial bone marrow disease detection by multiparameter flow cytometry in children with neuroblastoma. J Cancer Res Clin Oncol. 2019 Feb;145(2):535-542. doi: 10.1007/s00432-018-02831-w. Epub 2019 Jan 2. PMID: 30603901). This study was added as a reference to the material and methods - 2.3. Evaluation of Bone Marrow Samples section of our manuscript.
2# The authors present OS for flow cytometry positive and negative patients. They should present other clinical data of this population to support the accuracy of flow cytometry. For example, the stage, tumor burden of all patients. It is especially important to report those who had discordant results between flow cytometry and biopsy reports.
2# Unfortunately, due to the limited number of patients, it was not possible to create subgroups, including disease risk groups, for patients with positive and negative flow cytometry results.
2# Do they have have early stage disease and therefore, overdiagnosis by flow cytometry (MFC+BMB-) ? or they actually have advanced stage so bone marrow metastasis is very likely and only detected by flow cytometry?
2# The information on these patients were added to the results section.
3# The sample processing methods should also be described. Cancer cells may be lost either by processing the biopsy or flow cytometry samples and thereby affecting the accuracy of diagnostic tools.
3# The processing method was added to the Materials and Methods section - 2.3. Evaluation of Bone Marrow Samples
4# In total samples, N=16 (MFC+BMB-). Most of such samples are follow up samples so they are likely to have minimal residual disease, if MFC correctly detected the disease. It is very interesting. The authors should analyze the clinical outcome of this group in comparison with the MFC-BMB- group and MFC+BMB+ group. If the outcome is similar to the MFC+BMB+ group then MFC is a good tool in diagnosis of residual disease in marrow.
4# This is a very good point of view, however the results of follow-up bone marrow studies were also heterogeneous, unfortunately more patients and a longer follow-up period may reveal the relationship between MFC results and disease prognosis. MFC may be a good potential tool in diagnosis of residual disease in future with more studies.